# *Trichomonas vaginalis* Virus: Current Insights and Emerging Perspectives

**DOI:** 10.3390/v17070898

**Published:** 2025-06-26

**Authors:** Keonte J. Graves, Jan Novak, Christina A. Muzny

**Affiliations:** 1Division of Infectious Diseases, Department of Medicine, University of Alabama at Birmingham, Birmingham, AL 35294, USA; cmuzny@uabmc.edu; 2Department of Microbiology, University of Alabama at Birmingham, Birmingham, AL 35294, USA; jannovak@uab.edu

**Keywords:** TVV, *Trichomonas vaginalis* virus, *Trichomonasvirus*, *Totiviridae*, double-stranded RNA virus

## Abstract

*Trichomonas vaginalis*, a prevalent sexually transmitted protozoan parasite, is associated with adverse birth outcomes, increased risk of HIV and other sexually transmitted infections, infertility, and cervical cancer. Despite its widespread impact, trichomoniasis remains underdiagnosed and underreported globally. *Trichomonas vaginalis virus* (TVV), a double-stranded RNA (dsRNA) virus infecting *T. vaginalis*, could impact *T. vaginalis* pathogenicity. We provide an overview of TVV, including its genomic structure, transmission, impact on protein expression, role in 5-nitroimidazole drug susceptibility, and clinical significance. TVV is a ~5 kbp dsRNA virus enclosed within a viral capsid closely associated with the Golgi complex and plasma membrane of infected parasites. Hypothetical mechanisms of TVV transmission have been proposed. TVV affects protein expression in *T. vaginalis*, including cysteine proteases and surface antigens, thus impacting its virulence and ability to evade the immune system. Additionally, TVV may influence the sensitivity of *T. vaginalis* to treatment; clinical isolates of *T. vaginalis* not harboring TVV are more likely to be resistant to metronidazole. Clinically, TVV-positive *T. vaginalis* infections have been associated with a range in severity of genital signs and symptoms. Further research into interactions between *T. vaginalis* and TVV is essential in improving diagnosis, treatment, and the development of targeted interventions.

## 1. Introduction

*Trichomonas vaginalis*, the causative agent of trichomoniasis, is a parasitic protozoan that is estimated to be the most common non-viral sexually transmitted infection (STI) worldwide [1]. Globally, the burden of *T. vaginalis* infection is particularly high among women living in low-income areas and individuals aged 30–54 years [2]. There is also a predominant racial disparity, with non-Hispanic black women and men having a higher *T. vaginalis* prevalence than other groups of individuals [3]. *T. vaginalis* infection is associated with multiple adverse health outcomes, including adverse birth outcomes [4], increased risk of HIV and other STI acquisition [5,6], infertility [7], and increased risk of cervical neoplasia related to co-infection with human papilloma virus [8].

Despite its significant global burden of infection and association with multiple adverse outcomes, trichomoniasis is not a reportable disease and there is a limited public health response to this infection [9]. With the exception of HIV-infected women, screening is not routinely recommended in any population, although it should be considered in some high-risk populations (i.e., individuals presenting to STI clinics, incarcerated individuals, individuals with multiple sexual partners, those participating in transactional sex or engaging in drug misuse, and those with a history of STIs or incarceration) [10]. A significant proportion of infected individuals may be asymptomatic, up to 75% in some studies [11]. Among women who are symptomatic, a thin, frothy, yellow/green/gray malodorous vaginal discharge may be present. Additionally, women may also experience vulvar irritation, dysuria, or dyspareunia; less commonly, abdominal pain may be present [9]. *T. vaginalis* has occasionally been implicated in cases of pelvic inflammatory disease in women [12]. Symptomatic men may experience urethral discharge, a burning sensation or pain during urination or ejaculation, and/or itching of the urethra [13]. Men may also occasionally experience symptoms of epididymitis or prostatitis [14].

The diagnosis of trichomoniasis has traditionally relied on the wet mount microscopy of vaginal secretions in women in addition to culture in women and men [9]. However, these modalities have lower sensitivity compared to highly sensitive nucleic acid amplification tests (NAATs), which have flourished over the last decade [9]. *T. vaginalis* NAATs can be performed on both self-collected and clinician-collected genital specimens in women, as well as urine in women and men. Several *T. vaginalis* NAATs can be performed on demand at the point of care, leading to an accurate diagnosis and the treatment of the infection [9].

The treatment of *T. vaginalis* reduces signs and symptoms of infection, as well as decreasing sexual transmission to partners [10]. Oral 5-nitroimidazoles (metronidazole [MTZ], tinidazole [TDZ], and secnidazole [SEC]) are the only class of drugs approved by the FDA for the treatment of trichomoniasis [10]. Current guidelines recommend multi-dose oral MTZ (500 mg twice daily for 7 days) as the preferred treatment for trichomoniasis in women [10], including pregnant women, based on its superior efficacy in two multi-center randomized controlled trials (RCTs) [15,16]. In contrast, the recommended treatment for *T. vaginalis*-infected men is currently a single 2-g dose of oral MTZ [10], due to a lack of rigorous clinical trial data in men. A single 2-g dose of oral TDZ is an alternative treatment for both women and men [10]. A single 2-g dose of oral SEC has recently been approved by the FDA for the treatment of trichomoniasis in women and men after the current STI Treatment Guidelines were published, based on results from a recent RCT in women [17] and a review of the literature on prior studies in men [18]. 5-nitroimidazole drug resistance may occur among some *T. vaginalis* infections, posing a significant clinical challenge regarding treatment [19].

*T. vaginalis* can be divided into two distinct genetic population types: a more ancestral Type 1 population and a divergent Type 2 population; each population has unique phenotypes [20]. Both are well distributed globally; however, there are some differences. For example, the divergent Type 2 population has been associated with *T. vaginalis* parasites with higher mean minimum lethal concentrations to MTZ compared to the Type 1 population, suggesting that resistance to 5-nitroimidazoles may be more likely to develop in Type 2 parasites [20].

*Trichomonas vaginalis* virus (TVV) is a double-stranded RNA (dsRNA) virus, originally identified in 1985 [21,22], which can infect *T. vaginalis* and potentially contribute to its pathogenesis [23]. TVV has been found in the cytoplasm of *T. vaginalis*, encased within a viral protein capsid in the shape of a 120-subunit icosahedral (Figure 1) [24]. It is closely associated with the Golgi complex or adjacent to the plasma membrane [25]. TVV has traditionally been divided into four species (TVV1, TVV2, TVV3, and TVV4) that can co-infect *T. vaginalis* at the same time [26]; a fifth species (TVV5) has recently been identified [27]. In a recent systematic review and meta-analysis including 28 studies, the pooled global prevalence of TVV in *T. vaginalis* isolates was estimated to be 47% (95% CI, 39.3–54.8%); Brazil (90%) and South Africa (82%) had the highest prevalence [28]. Interestingly, the ancestral Type 1 population of *T. vaginalis* has been associated with more frequent TVV infections [20].

The purpose of this article is to provide an update on TVV, including data on its genomic structure, transmission methods, impact on *T. vaginalis* protein expression, role in 5-nitroimidazole drug susceptibility, and overall clinical significance.

## 2. Genetic Structure and Transmission of TVV

Virus-like particles were initially observed in the cytoplasm and closely associated with the plasma membrane and Golgi complex of infected *T. vaginalis* strains, as reported in the mid-1980s [22,29,30,31]. These initial observations subsequently led to the discovery of the 4.5–5 kilobase pair (kbp) dsRNA characteristic of the *Trichomonasvirus* genus and TVV species [21,26,32,33].

Until recently, only four species of TVV (TVV1, TVV2, TVV3, and TVV4) had been described within the *Trichomonasvirus* genus [23]. However, a 2022 study characterized at least six novel strains belonging to a fifth TVV species, named TVV5 [27]. Using a technique known as transcriptomic mining, Manny et al. were able to screen RNA-Seq transcriptomic data deposited from various studies of the gene sequences of the initial four species of TVV [27]. This uncovered the existence of the novel TVV5 species. The discovery of this novel TVV species will require additional investigation into its prevalence and distribution among *T. vaginalis* isolates.

Similarly to the other dsRNA viruses found in the *Totiviridae* family of viruses, the TVV genome consists of two overlapping open reading frames, *GAG* and *POL*, encoding a viral capsid protein and RNA-dependent RNA polymerase (RDRP) (Figure 1), respectively [34]. The viral capsid that contains the dsRNA genome is arranged in an icosahedral shape that possesses deep grooves/channels that have been hypothesized to aid in the transmission of the virus (Figure 1) [24].

The mechanisms of TVV transmission from an infected trichomonad to an un-infected trichomonad are still not well understood. There have been several transmission mechanisms proposed since the discovery of virus-like particles within the cytoplasm of *T. vaginalis* parasites [22,30]. A hypothetical mode of TVV transmission is theorized to take place through vertical transmission, during the parasite’s replication. *T. vaginalis* primarily replicates through longitudinal binary fission [14,35]. This would allow for the transmission of TVV from a TVV-infected parent trichomonad to the “daughter” trichomonads. Another theorized strategy involves transmission through small extracellular vesicles (sEVs) [36]. As virus-like particles and TVV virions have been observed to be closely associated with the Golgi complex and plasma membranes of *T. vaginalis*, their presence within sEVs is to be expected (see Section 3.3, below). These sEVs can be taken up by uninfected *T. vaginalis* parasites through endocytosis, potentially mediating dsRNA virus transmission [36]. However, future studies are needed to evaluate the significance of this pathway and its potential clinical impact.

## 3. Impact of TVV on *T. vaginalis* Protein Expression

### 3.1. Cysteine Proteases

*T. vaginalis* infection involves the adherence of the protozoan to host cells followed by host cell lysis. The *T. vaginalis* genome has >400 protease genes, and many of the encoded proteases are thought to participate in the latter process [37,38]. These proteases, such as cysteine proteinase TvCP2, are overexpressed under glucose- and/or iron-limited conditions [39,40,41,42]. Various studies have investigated the effects of TVV on protein expression in *T. vaginalis* isolates; the findings have varied, showing either some effects or no impact on protein expression [43]. Despite this, there is evidence that TVV affects the expression of various proteases, including cysteine proteinases [44,45]. These proteases are secreted using the lysosomal pathway wherein glycosylation enables lysosomal protein targeting [46]. Thus, a combination of nutritional stress and the presence of the endosymbiont TVV can elevate the production of *T. vaginalis* proteases that in turn enhance host–cell adherence and cytotoxicity and may mediate the evasion of the host immune response.

### 3.2. Surface Antigens

*T. vaginalis* protein P270 is a ~270-kDa cell-surface protein known for its immunogenic properties [45,47,48,49,50]. This protein has variable expression, described as a phenotypic variation of P270 that occurs in only *T. vaginalis* isolates infected with TVV [51,52]. Comparative studies of distinct T. vaginalis isolates without TVV and with TVV have revealed differential cell localizations of P270 in these isolates. TVV-negative isolates only expressed P270 in the cytoplasm, whereas TVV-positive isolates had mixed cell subpopulations with surface and intracellular P270 that exhibited phenotypic variation [53]. A follow-up study revealed that this compartmentalization and phenotypic variation in surface localization in P270 depends on the presence of TVV [50]. An important question to answer in future studies is whether the TVV-mediated surface localization of P270 impacts the pathogenicity of the isolates infected with TVV. Further studies will also need to define the potential of the five known TVV species to affect the surface localization of P270 and define the mechanisms involved in the process to delineate potential implications for clinical care for patients with trichomoniasis.

### 3.3. Other Proteins

*T. vaginalis* can produce multiple virulence-factor proteins that mediate interactions with host cells in the vaginal environment, as well as the vaginal microbiota, and promote its pathogenetic processes. Some of these proteins can be cell-associated, whereas others can be secreted, either utilizing the endoplasmic reticulum–Golgi complex secretory pathway or pathways using lysosomes or extracellular microvesicles [46,54,55,56,57,58]. *T. vaginalis* generates two types of extracellular microvesicles: small microvesicles termed exosomes (50–150 nm in diameter) and larger microvesicles (>200 nm in diameter) [46,59]. The larger microvesicles are derived from the cell–plasma membrane in a budding process [56,60], whereas exosomes form intracellularly in the lumen of endosomal multivesicular bodies [59]. Both types of extracellular vesicles contain proteins and RNA and can also contain TVV virions [61]. These extracellular vesicles can be taken up by the host cells and exert immunomodulatory effects. The vesicles can also mediate adherence to the host cells to aid in the pathogenesis of *T. vaginalis* infection [56,57,60].

Recent studies have provided new information about the clinical relevance of TVV and exosomes in *T. vaginalis* infection [36,58]. Specifically, TVV can be exported together with cellular proteins and RNA in the exosomes to the extracellular environment. Experiments using TVV-positive and TVV-negative *T. vaginalis* isolates revealed differences in the protein and RNA cargo of the respective exosomes. Furthermore, exosomes from TVV-positive *T. vaginalis* isolates induced higher pro-inflammatory responses in cultured epithelial cells compared to those from TVV-negative clones. Together, these results suggest that TVV-positive *T. vaginalis* isolates have an elevated potential to induce pathogenic inflammatory responses in the infected host [58]. As there are over 200 proteins in the *T. vaginalis* exosomes, future studies should identify those with the most pathogenic potential as possible therapeutic targets.

## 4. 5-Nitroimidazole Drug Susceptibility

The 5-nitroimidazole class of drugs, consisting of MTZ, TDZ, and SEC, are the only drugs approved by the FDA for the treatment of *T. vaginalis* infections in the United States. There have been a limited number of studies investigating associations between the presence of TVV species and metronidazole resistance [62,63,64,65,66]. To date, there is no correlation between the presence of TVV and 5-nitroimidazole resistance in infected *T. vaginalis* parasites. In contrast, two studies found that clinical isolates of *T. vaginalis* not harboring TVV are more likely to be resistant to metronidazole [63,65]. Further investigation is needed to elucidate whether this relationship is observed for other 5-nitroimidazoles (i.e., TDZ and SEC) used for treatment.

## 5. Clinical Significance

Multiple studies have investigated the clinical significance of TVV infecting *T. vaginalis* isolates [25,67,68,69,70,71], particularly because its presence influences the total protein expression of *T. vaginalis*. One such study found evidence that host (human) cells were able to sense the TVV dsRNA of infected trichomonads through the host cell’s Toll-like receptor 3 (TLR3), which is part of the innate immune system [72]. Notably, TLR3 is primarily responsible for recognizing and detecting viral dsRNA. TLR3 activates a host immune response that induces the production of pro-inflammatory interferons and cytokines.

Few studies have reported an association between TVV presence and symptomatic *T. vaginalis* infections [69,70]. In addition, specific signs and symptoms experienced by *T. vaginalis*-infected patients have been associated with the presence of specific TVV species. For instance, mild to moderate symptomatic infections have been associated with TVV1 [25]. On the other hand, moderate-to-severe symptomatic infections have been associated with TVV2 [25]. The most common complaints experienced by patients with symptomatic *T. vaginalis* infections positive for TVV have included vaginal discharge, dysuria, and vaginal erythema [25,70]. Additional studies are needed to characterize the role that other TVV species (TVV3, TVV4, and TVV5) may play in the severity of symptoms among individuals with symptomatic *T. vaginalis* infections.

## 6. Conclusions and Future Directions

In conclusion, *T. vaginalis* remains a significant global health concern, with a high burden of infection, particularly among women in low-income areas and specific high-risk populations. The discovery of TVV adds an additional layer of complexity to the pathogenesis of trichomoniasis, particularly considering the impact on *T. vaginalis* protein expression and its potential influence on drug susceptibility. While a better understanding of the exact role that TVV plays in the pathogenesis of trichomoniasis is needed, emerging evidence suggests that TVV may play a crucial role in modifying the virulence and immune escape of *T. vaginalis* in the infected individuals. To date, the majority of clinical TVV research has been performed using *T. vaginalis* isolates obtained from *T. vaginalis*-infected women. Future studies are needed to investigate the role of TVV in *T. vaginalis*-infected men. Additionally, the potential correlation between TVV presence and an increased sensitivity to 5-nitroimidazoles warrants further exploration. Understanding the interplay between *T. vaginalis* and TVV is crucial in advancing the clinical management of trichomoniasis, including the development of improved diagnostic, therapeutic, and preventive strategies. Future research will be essential in elucidating the overall clinical implications of TVV and its potential as a target for novel therapeutic interventions.

## Figures and Tables

**Figure 1 viruses-17-00898-f001:**
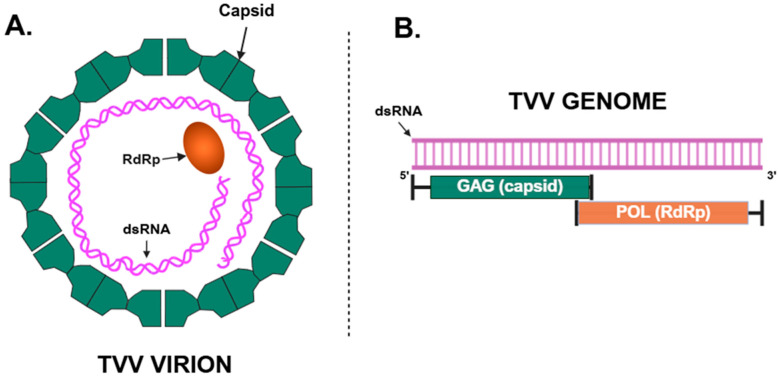
*Trichomonas vaginalis* virus (TVV) virion and genome. (**A**) TVV Virion: the dsRNA genome [*pink*] of TVV enclosed within a viral capsid [*green*]. (**B**) TVV Genome: the TVV genome consists of two overlapping open reading frames, GAG [*green*] and POL [*orange*], which encode the viral capsid and RdRp, respectively. Abbreviations: RdRp, RNA-dependent RNA polymerase; dsRNA, double-stranded RNA.

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
