# Peer review of "Trichomonas vaginalis Virus: Current Insights and Emerging Perspectives"

_viruses, 2025, doi:10.3390/v17070898_

Round 1
Reviewer 1 Report
Comments and Suggestions for Authors
In this manuscript, the authors review previous studies on Trichomonas vaginalis virus (TVV), including its virology and potential effects on the parasite host. However, most of the points discussed have already been reported in the literature. For example:
- Trichomonasvirus: a new genus of protozoan viruses in the family Totiviridae [DOI: 10.1007/s00705-010-0832-8]
- Trichomonas vaginalis virus: a review of the literature [DOI: 10.1177/0956462418809767]
Notably, the second reference was published by the same authors as this manuscript, and there appears to be little new information compared to their previous work. Several specific concerns are outlined below:
Lines 18 & 128–137
The manuscript states that TVV can be transmitted via cell division or through small extracellular vesicles. However, this has not been experimentally confirmed. The authors should revise this section to clarify that the transmission mechanisms of TVV remain uncertain, and that vertical transmission is only a proposed hypothesis based on limited evidence.
Lines 18–25 and Section 3 (starting at Line 138)
The authors suggest that TVV influences virulence, drug resistance, and protein expression in T. vaginalis. However, several studies have reported no such associations. To provide a more balanced view, the authors should acknowledge these contradictory findings and offer a discussion on possible reasons for the discrepancies.
Line 109
The text implies that satellite RNAs are part of the viral genome. However, in the cited reference, the satellite RNAs were found in virus-positive isolates but are not components of the viral dsRNA genome. The authors should revise this statement to clarify that these RNAs may act as auxiliary elements necessary for replication, rather than being part of the viral genome.
Line 172
The manuscript refers to vesicles larger than 200 nm as “large vesicles,” but the correct term is “microvesicles.” Furthermore, the cited reference does not define or mention microvesicles. The terminology should be corrected for accuracy, and the appropriateness of the citation should be re-evaluated.
Reviewer 2 Report
Comments and Suggestions for Authors
This review provides an excellent summary of the current knowledge regarding TVVs and their role in the pathogenesis of Trichomonas vaginalis.
There are few minor comments related with the manuscript which I list below:
1) Fig 1 is slightly blurry.
2) Line 114: The dot after "Manny et al" is missing
3) Lines 140, 141,147,151,153 and 154: T. vaginalis is not in italics
4) Line 161: please change the word "strain" to "isolate"
5) Line 186: the sentence can be a bit confusing for the reader. I recommend simplifying it by using an expression such as ‘TVV+ isolates’ or ‘parasites/trichomonads harboring TVV’.
6) Line 207: specify that those interferons and cytokines are proinflammatory
7) Regarding the paragraph related with simptoms and TVV, only Fraga et al., 2012 have reported that association . I recommend to highlight this aspect.
Round 2
Reviewer 1 Report
Comments and Suggestions for Authors
The revised version is acceptable for publication